# Competitiveness and Phylogenetic Relationship of Rhizobial Strains with Different Symbiotic Efficiency in *Trifolium repens*: Conversion of Parasitic into Non-Parasitic Rhizobia by Natural Symbiotic Gene Transfer

**DOI:** 10.3390/biology12020243

**Published:** 2023-02-03

**Authors:** María A. Morel Revetria, Andrés Berais-Rubio, Matías Giménez, Juan Sanjuán, Santiago Signorelli, Jorge Monza

**Affiliations:** 1Laboratorio de Bioquímica, Facultad de Agronomía, Universidad de la República, Av. Garzón 780, Montevideo 11300, Uruguay; 2Laboratorio de Microbiología de Suelos, Facultad de Ciencias, Universidad de la República, Igua 4225, Montevideo 11400, Uruguay; 3Laboratorio Microbiología Molecular, Departamento BIOGEM, Instituto de Investigaciones Biológicas Clemente Estable (IIBCE), Av. Italia 3318, Montevideo 11600, Uruguay; 4Laboratorio de Genómica Microbiana, Instituto Pasteur, Mataojo 2020, Montevideo 11400, Uruguay; 5Departamento de Microbiología del Suelo y la Planta, Estación Experimental del Zaidín, Consejo Superior de Investigaciones Científicas (CSIC), E-18008 Granada, Spain; 6Plant Energy Biology, School of Molecular Science, The University of Western Australia, Crawley, WA 6009, Australia

**Keywords:** biological nitrogen fixation, symbiosis, inoculants, competitiveness, parasitic strains, gene transfer, *hrrP* gene, *sapA* gene

## Abstract

**Simple Summary:**

Partially efficient and inefficient rhizobia strains can decrease biological nitrogen fixation in legumes, especially when not inoculated. This work shows the high nodulation competitiveness in *Trifolium repens* of two strains with intermediate nitrogen fixation capacity. The genome sequence of one of them resulted in the first identification of *Rhizobium redzepovicii* as a *T. repens* symbiont. The analysis of symbiotic genes of different Rhizobium sp. capable of nodulating *T. repens* evidences the horizontal transfer of genes from efficient to inefficient strains, leading to a strain with intermediate efficiency. For the first time, sequencing, and analysis of the genome of the rhizobium used as clover inoculant in Uruguay confirmed that it is a strain belonging to *R. leguminosarum* phylogenetically close to bv. viciae. The search for *hrrP*-like genes in different Rhizobium species shows their presence in the chromosome and plasmids. This is interesting because *hrrP* encodes a metallopeptidase of the M16A family whose activity is involved in symbiotic efficiency. Our results highlight the need for farmers to make an effort in seed inoculation to improve clover yields and mitigate the effect of parasitic and low-efficiency strains present in soils.

**Abstract:**

In Uruguayan soils, populations of native and naturalized rhizobia nodulate white clover. These populations include efficient rhizobia but also parasitic strains, which compete for nodule occupancy and hinder optimal nitrogen fixation by the grassland. Nodulation competitiveness assays using *gusA*-tagged strains proved a high nodule occupancy by the inoculant strain U204, but this was lower than the strains with intermediate efficiencies, U268 and U1116. Clover biomass production only decreased when the parasitic strain UP3 was in a 99:1 ratio with U204, but not when UP3 was at equal or lower numbers than U204. Based on phylogenetic analyses, strains with different efficiencies did not cluster together, and U1116 grouped with the parasitic strains. Our results suggest symbiotic gene transfer from an effective strain to U1116, thereby improving its symbiotic efficiency. Genome sequencing of U268 and U204 strains allowed us to assign them to species *Rhizobium redzepovicii*, the first report of this species nodulating clover, and *Rhizobium leguminosarun*, respectively. We also report the presence of *hrrP*- and *sapA*-like genes in the genomes of WSM597, U204, and U268 strains, which are related to symbiotic efficiency in rhizobia. Interestingly, we report here chromosomally located *hrrP*-like genes.

## 1. Introduction

Red (*Trifolium pratense*) and white (*T. repens*) clovers were introduced in Uruguay during the 1930s to improve pastures. Between 1964 and 1967, both clovers were inoculated with a mixture of rhizobial strains U20 and U26 (=TA1); and later, *R. leguminosarum* bv. trifolii strain U204 from the USA was the selected inoculant [1].

In soils in which clovers are sown, there are populations of native-naturalized clover rhizobia, which include strains with similar symbiotic efficiency to the commercial inoculant U204, other strains with lower efficiency [2,3], and even so-called parasitic rhizobia that are ineffective [4,5,6]. Moreover, analysis of 16S rRNA and ITS sequences, as well as housekeeping (*atpD*, *glnII*, *recA*, *rpoB*) and symbiotic genes (*nodA*, *nodC*, *nifH*), allowed us to determine the identities and phylogenetic relationships among these rhizobia [7]. The native parasitic clover rhizobia are phylogenetically distinct from the efficient rhizobia, either native to Uruguay or introduced in the country [7].

Clover parasitic rhizobia were isolated from *T. polymorphum* [6,7] and *T. pallidum* [4]. These rhizobia are Fix^-^ in *T. repens*, *T. pratense*, other perennial clovers from North America and Africa, and annual clovers from the Mediterranean region [4,8]. Conversely, rhizobia isolated from *T. pratense* and *T. repens* are generally Fix^-^ in *T. polymorphum* [7].

Nodulation of clovers by Fix^-^ rhizobia hinders the successful implantation of clover pastures, especially those where seeds are not inoculated with selected efficient rhizobia and in fields with no previous history of clover [1,3]. Although this situation is known to be risky and negative for pasture success, there is a lack of fundamental knowledge on the nodulation competitiveness of parasitic rhizobia and the causes of their inefficiency.

Crook et al. [9] demonstrated that the loss of HR (host range) rhizobial accessory plasmids gave rise to strains with high symbiotic efficiency in *Ensifer meliloti*–*Medicago truncatula* symbiosis. The opposite was also true: the acquisition of HR plasmids could transform an efficient strain of *E. meliloti* into an inefficient one [9]. The *hrrP* (host range restriction peptidase) gene was reported in these plasmids; *hrrP* encodes a peptidase of the M16 family [10] that hydrolyses NCR (nodule cysteine-rich) peptides produced by the host [11]. Given that NCR peptides are necessary for nodule development, *hrrP* expression was associated with the inefficient symbiotic phenotype [10]. In turn, the overexpression of a M16 peptidase encoded by the *sapA* chromosomic gene leads to discrete symbiotic deficiencies [12].

In *Trifolium* spp., the host–rhizobia incompatibility/compatibility was related to diversity centers and annual or perennial lifecycles of clovers [13,14]. Thus, it seems unlikely that rhizobia associated with native clovers can cross those geographical and phenological barriers to establishing efficient symbiosis with introduced clovers, such as *T. repens* and *T. pratense* in Uruguay. However, bacterial symbiotic genes are usually located in highly mobile genetic elements (such as symbiotic islands and plasmids) and can be horizontally transferred between strains from different species and even genera [15,16]. Horizontal gene transfer (HGT) events can transform non-symbiotic rhizobia in soils in which a legume grows into a new symbiont of that legume host [15,17]. Tartaglia et al. [7] reported evidence of HGT in certain clover rhizobia strains and proposed that two newly isolated, competitive, and efficient clover strains could be descendants of native rhizobia that have acquired symbiotic genes, probably donated by the inoculant strain used in Uruguay since 1967. These newly arisen symbiotic rhizobia were shown to be phylogenetically related to the inoculant strain. On the contrary, these authors could not detect HGT of symbiotic genes from the inoculant to parasitic clover rhizobia, which are phylogenetically more distant from the inoculant.

In this context, we have genetically characterized autochthonous strains from Uruguayan soils displaying different symbiotic efficiencies in white clover, evaluated their nodulation competitiveness, and asked whether these rhizobia have enough genetic plasticity for their adaptation to introduced clovers.

## 2. Material and Methods

Rhizobial strains UP3 (=P3), U1116 (=1116), U268 (=268), and U314 (=314) were isolated from Uruguayan soils. The UP3 strain was previously identified in our group as *R. acidisoli* [7]. The *R. leguminosarum* bv. trifolii U204 strain (USA, Nitragin, =CIAT 2445 = U28) was obtained from the Biochemistry Laboratory rhizobia collection (Facultad de Agronomía, Universidad de la República) (Table 1).

Rhizobia were routinely grown at 28 °C in yeast extract-mannitol [18]. *E. coli* strain S17-1 ʎ -pir containing the plasmid pCAM131 that carries the transposon mTn5SS*gusA*31 [19] was grown in LB medium [20] at 37 °C supplemented with spectinomycin 100 µg/mL and streptomycin 100 µg/mL. Bacteria were preserved at 4 °C in a solid medium or at −80 °C in 12% glycerol in the respective medium [21].

### 2.1. Plant Assays

Seeds of *T. repens* cv. “Estanzuela Zapicán” were surface sterilized and germinated as described by Howieson and Dilworth [21]. Briefly, seeds were treated with 90% (*v*/*v*) ethanol for 1 min, followed by 4 min in 20% (*v*/*v*) NaClO, then rinsed three times in sterile distilled water. Seeds were spread on sterilized water agar plates and incubated at 25 °C in the dark until the radicles fully emerged. Seedlings were transferred to 350 mL pots containing sterile vermiculite:sand (1:1). Each seedling was inoculated with 100 μL of a rhizobial suspension at the densities indicated below for each test. Each treatment included five pots with five plants/pot. The plants were cultivated in a growth chamber at 23/20 °C (day/night), with a 16/8 h photoperiod and 220 mE m^2^. s^−1^ photosynthetic photon flux density.

#### 2.1.1. Symbiotic Efficiency Tests

Each seedling was inoculated with approximately 10^6^ colony-forming units (CFUs) of individual rhizobial strains per seed. Uninoculated plants with and without nitrogen were the control treatments (+N or −N, respectively). Plants were irrigated every three days with water or Hornum medium without nitrogen [22]. +N control plants were rinsed with water or Hornum medium supplemented with KNO_3_ (5 mM). Plants were harvested at 35- and 60- days post inoculation (dpi), and the shoot dry weight (SDW) was determined after drying at 70 °C for 72 h. Plant SDW accumulated in both plants’ harvests was used to determine the symbiotic efficiency.

#### 2.1.2. Nodulation Competitiveness Tests

To test the nodule occupancy by rhizobia, U204 and UP3 strains were tagged with the reporter gene *gusA* using the *Escherichia coli* strain S17-1 ʎ-pir, as described in [2]. The nodulation kinetics and symbiotic efficiency of a few transconjugants were evaluated in *T. repens*, as described previously [3]. UP3::*gusA* and U204::*gusA*, showing a symbiotic behavior similar to the untagged wild types UP3 and U204, were selected for the nodulation competitiveness assays under controlled conditions. CFU/seed were counted as described in Riviezzi et al. [23].

Competitiveness Test 1. Each seedling was inoculated with mixtures of U204::*gusA*/UP3, U204::*gusA*/U268, or U204::*gusA*/U1116 strains in 1:1 ratios. Control plants were inoculated with each strain individually. The CFU per seed of U204::*gusA*, UP3, U268, and U1116 strains were 4 × 10^6^, 5 × 10^6^, 3 × 10^6^, 5 × 10^6^, respectively. Plant growth conditions were as described above.

Competitiveness Test 2. Each seedling was inoculated with UP3::*gusA*, U204, or mixtures of both, as detailed in Table 2. Treatments M1, M2, and M3 indicated in Table 2 correspond to mixtures of UP3::*gusA* and U204 strains in a ratio of 99 to 1, 1 to 99, and 1 to 1 (50% of each), respectively. The final concentrations determined are shown in Table 2. Plant growth conditions were as described above.

In both competitiveness tests, at 21 dpi, the roots were collected, washed with distilled water and phosphate buffer (PBS), placed in tubes with a solution containing 1% SDS, 0.5 M EDTA, and X-Gluc (Thermo Scientific™, R0851) in PBS 50 mM pH 7.0, and incubated at 37 °C in the dark. The plant SDW was recorded for each treatment. Total nodules were recorded, distinguishing those occupied by U204::*gusA* or UP3::*gusA*, which turned blue owing to the expression of β-glucuronidase activity.

### 2.2. Rhizobia Genomic DNA Extraction, Amplification, and Sequencing

Fresh cultures of strains U204 and U268 were used for the genomic DNA extraction with a DNA extraction kit (Qiagen, Hilden, Germany). DNA quality and concentration were determined with a *Nanodrop 2000* (Thermo Scientific, Waltham, MA, USA), and its integrity was visualized after electrophoresis in 1.2% agarose gel.

#### 2.2.1. Amplification and Sequencing of ITS Region, 16S rRNA, Housekeeping, and Symbiotic Genes

Internal transcribed spacer (ITS) 16–23S rRNA was PCR amplified using primers ITS322 and ITS340 [24]. An inner region of the 16S ribosomal RNA (rRNA) sequence was amplified using the universal primers 27f and 1525r [25]. The housekeeping genes *atpD*, *glnII*, and *recA* were amplified using the primers atpD273f-atpD771r [26], glnII12F-glnII689R, and recA41F-recA640R [27], respectively. The symbiotic genes *nodA*, *nifH*, and *nodC* were amplified using the primers nodA3F-nodA4R [28], *nifHI–nifHF*, and *nodCF*, *nodCFn* and *nodCI*, respectively [29]. The conditions used for PCR amplification were those described by the authors, with a modification for *nodA* as described by Tartaglia et al. [7]. PCR products were run at 100 V in Tris-acetate buffer pH 8.2 in 1.2% agarose gels, with 1-kb ladder (Maestro) and stained with SYBR™ Safe DNA Gel Stain. Gel images were captured using the Kodak Gel Logic 100 Imaging System. The PCR products were sequenced by Macrogen (Macrogen, Inc., Seoul, Republic of Korea).

#### 2.2.2. Phylogenies of ITS, 16S rRNA, Housekeeping, and Symbiotic Genes

The sequences were analyzed by comparison with reference strains available in GenBank (http://www.ncbi.nlm.nih.gov, accessed on 15 November 2022). Low-quality bases were removed using Chromas v. 2.6.4. Sequence alignment with the Clustal W algorithm and phylogenetic tree constructions (maximum likelihood) were performed using MEGA 11 software [30]. Statistical support for tree nodes was evaluated by bootstrap analyses using 1000 replicates [31].

GenBank accession numbers for partial nucleotide sequences from strains U1116, U268, and U314 are MZ393150, MZ393151, and MZ393152 for *rRNA* 16S gene; MZ401359, MZ401357, and MZ401358, for the ITS region; MZ401104, MZ401102, and MZ401103 for the *rpoB* gene; or MZ401101, MZ401099, and MZ401100 for the *recA* gene; MZ401088, MZ401089, and MZ401087 for the *glnII* gene; or MZ401086, MZ401084, and MZ401085 for the *atpD* gene; MZ401095, MZ401093, and MZ401094 for *nodA;* MZ401096, MZ401097, and MZ401098 for *nodC*; and MZ401092, MZ401090, and MZ401091 for the *nifH* gene, respectively.

#### 2.2.3. U204 and U268 Whole-Genome Sequencing

The genomic library preparation and its further sequencing were performed by Macrogen (Seoul, Republic of Korea). Genomic sequencing was performed with Illumina Technology (HiSeq 2500 system), yielding paired-end reads with a length of 151 bp. Read quality was checked with FastQC (v.0.11.6) and then filtered with Trimmomatic (v.0.38) software [32]. De novo genome assembly was performed with SPAdes software. Genomic annotation of U204 and U268 genomes was achieved using the Rapid Annotations using Subsystems Technology (RAST) [33].

The Whole Genome Shotgun projects for U204 and U268 have been deposited at DDBJ/ENA/GenBank under the accessions JAPMMB000000000 and JAPPSO000000000, respectively. The versions described in this paper are version JAPMMB010000000 and JAPPSO010000000. The genome sequences of strains WSM597 and WSM2304 were downloaded from GenBank under the RefSeq assembly accessions GCF_000271785.1 and GCF_000021345.1, respectively.

The predicted genes were functionally categorized using the SEED subsystems [34] at the RAST server. The calculation of the Average Nucleotide Index (ANI), correlation indexes of tetra-nucleotide signatures, and tetra correlation search (TCS) were performed using the JSpeciesWS server [35]. The genome sequence data were uploaded to the Type Genome Server (TYGS) for a whole genome-based taxonomic analysis [36].

The comparison and annotation of orthologous gene clusters among U204, U268, and WSM597 were achieved using the web platform OrthoVenn (OrthoVenn2, https://orthovenn2.bioinfotoolkits.net, accessed on 9 December 2022). Putative M16 peptidases were searched in WSM597, U204, and U268 genomes by BlastP against reference Hrrp (GenBank ID: AJT61688.1) SapA (GenBank ID: CAC45492.2) aminoacidic sequences of *E. meliloti*. The detection of plasmidic locations of the coding genes was analyzed by plaSquid [37]. Interpro was used to scan for M16 peptidase domains [38].

### 2.3. Statistical Analysis

Post-hoc pairwise comparison was analyzed from plant assays based on LSD (least significant difference) with ANOVA to explore differences between means. The statistical significance was determined at *p <* 0.05. The values shown in the tables represent the mean ± standard error of the mean. Homogeneous groups were designated by letters. The analysis was performed using the InfoStat package [39].

## 3. Results

### 3.1. Symbiotic Efficiency of Autochthonous Strains on White Clover

The symbiotic efficiency of autochthonous strains U314, U268, and U1116 on *T. repens* was determined by comparing plants’ accumulated shoot dry weight (SDW) after two consecutive harvests. The symbiosis was considered efficient when the SDW of inoculated plants was >70% of the SDW of plants fertilized with KNO_3_ (control +N). The symbiosis was considered inefficient when the plant SDW was the same as the −N control plants. Thus, the commercial inoculant strain U204 and the parasitic strain UP3 established efficient and inefficient nitrogen-fixing symbioses, respectively, and strains U268, U314, and U1116 had intermediate symbiotic efficiencies (Table 3).

### 3.2. Nodulation Competitiveness of Efficient, Intermediate, and Parasitic Strains in White Clover

In Test 1, the competitiveness of each strain was established by the capacity to induce the formation of nodules in the roots of T. repens when the seeds were co-inoculated with the commercial inoculant (U204::*gusA*) at 1:1 ratios. No differences were found in terms of U204 occupation relative to any of the other three strains tested (Table 4). Shoot biomass at 22 days post inoculation (dpi) was the same when plants were inoculated with the different mixtures of strains tested or solely inoculated with U268 and U1116, but lower when inoculated with UP3 (Table 4). Regarding nodulation, the total number of nodules per plant was the same in all treatments (Table 4).

The nodule occupancy by U204::*gusA* was higher than UP3 but lower than those of the U1116 and U268 strains, which occupied 58 and 75% of nodules, respectively, when mixed with U204::*gusA* (Table 4). It is worthwhile to remark that the nodules formed by strain UP3 were tiny and white (see later on), so we could have underestimated them. Therefore, we used the UP3 strain tagged with *gusA* gene (UP3::*gusA*) in competitiveness test 2, which allowed us to improve the counting of parasitic nodules.

In competitiveness test 2, we used different ratios in the mixtures, including strains U204 and UP3::*gusA*. The parasitic strain UP3::*gusA* was detectable in all treatments in which it was included in the inoculation mixture. The treatments including a high percentage of UP3 (i.e., single inoculation with strain UP3::*gusA* and co-inoculation with 100 to 1 of UP3::*gusA* to strain U204 -M2 treatment-) led to the highest number of nodules per plant at 21 dpi. Moreover, in M2 treatment, most nodules were infected with UP3::*gusA*. Conversely, when U204 was the most abundant strain (M1), the occupancy by UP3::*gusA* was reduced. However, with the ratio 1:1 (UP3::*gusA*/U204, M3), the UP3 nodule occupancy was much higher than U204 (Table 5).

In terms of biomass production, treatments including UP3::*gusA* alone or at a high ratio (M2 treatment) led to lower plant biomass production, compared to all other treatments (Table 5).

It seems likely that the different nodule occupancies displayed by strain U204 and UP3 between Table 4 and Table 5 is a consequence of underestimating the number of nodules induced by UP3 (Figure 1A,B). The efficient strain U204 induced cylindrical, elongated, typically indeterminate-type nodules (Figure 1A). These nodules were of a pinkish color, indicative of the presence of leghaemoglobin, and they were mostly located in the main root (Figure 1B). On the other hand, root nodules induced by UP3 were spherical in shape, of small size, and white (Figure 1A), and they predominated in lateral roots. The presence of mixed nodules (partially stained by strain UP3::*gusA*) on roots was also observed (Figure 1B).

### 3.3. Phylogenetic Relationship between Rhizobia with Different Symbiotic Efficiencies

Sequences of the 16S rRNA gene from strains U268, U314, and U1116 presented 99.9 to 100% of identity with sequences of *Rhizobium* sp. (Appendix A). The analysis of the ITS region sequences showed that strain U1116 grouped with UP3, UP33, WSM597, and WSM2304, all native strains, was inefficient in *T. repens*. On the other hand, strains U314 and U268 clustered together in a distinctive branch separated from all the other strains (Figure 2).

A multilocus sequence analysis (MLSA) was performed with housekeeping and symbiotic genes. We analyzed the phylogenetic relationship of housekeeping genes *atpD* (405 bp), *glnII* (416 bp), and *recA* (375 bp) and the symbiotic genes *nodA* (387 bp), *nodC* (482 bp), and *nifH* (317 bp). In the phylogenetic tree derived from the MLSA of the concatenated housekeeping genes, the strains U1116, U268, U314, and UP3 were distributed into two distinct clusters (Figure 3). The analysis showed that strain U1116 grouped together with the parasitic strains UP3, UP33, WSM597, and WSM2304. Once again, strains U314 and U268 were grouped separately from all other strains (Figure 3). U204 was in a clade that included different clover-efficient strains, such as strains U249, UN2, and U317 (Figure 3).

The maximum likelihood tree based on the symbiotic gene sequences separated the strains into clover-efficient and -inefficient rhizobia strains. The parasitic strains were clustered in one well-supported branch. In contrast, the strains with intermediate and low efficiency (U314, U268, and U1116) clustered together in a large branch comprising the efficient nitrogen-fixing strains that included strains used in commercial inoculants, such as U204 and TA1 (Figure 4).

### 3.4. Draft Genome of U204 and U268 Strains

The genome of the commercial strain U204 was unknown; therefore, we sequenced it. The sequenced genome is 7.69 Mb in size and 60.6% GC content. It contains 7957 coding sequences and 51 RNA genes. Based on TYGS analysis, the U204 strain belongs to the species *Rhizobium leguminosarum* (Appendix A). Among *R. leguminosarum*, U204 shared z-scores > 0.999 with *R. leguminosarum* bv. viciae and *R. leguminosarum* bv. trifolium in TCS and tetra-nucleotide signature. Moreover, the U204 genome shared ANIb and ANIm values ≥ 93% with both bv. of *R. leguminosarum*, as well as ANIb and ANIm values ≥ 96% with *R. leguminosarum* bv. viciae (Appendix A). Among the 18 distinct genospecies groups by Young et al. [40] for the *Rhizobium leguminosarum* species complex, the strain U204 belongs to the genospecies group E: *R. leguminosarum* (Appendix A).

Since U268 was separated from other species in the phylogenetic analysis of ITS and housekeeping genes (Figure 2 and Figure 3), we obtained its genome sequence as well. The U268 draft genome revealed that it is 6.87 Mb in size. With 61.2% GC content, it contains 6998 coding sequences and 48 RNA genes. Based on TCS, several *Rhizobium* spp., including *R. chutanense*, *R. leguminosarum* bv. trifolii, *R. hidalgonense*, *R. ecuadorense*, *R. redzepovicii*, and *R. acidisoli*, were found to be related with the U268 strain considering z-scores > 0.999; however, based on the tetra-nucleotide signature, only *R. redzepovicii* and *R. acidisoli* showed z-scores > 0.999. Further, we found ANIb and ANIm values ≥ 96% only when comparing the U268 genome with *R. redzepovicii*, which consistently identified U268 as *R. redzepovicii*. The phylogenetic analysis of *R. redzepovicii* U268, *R. redzepovicii* 18T, and other *Rhizobium* spp. strains showed that the 18T and U268 strains genomes cluster together (Figure 5), and according to TYGS analysis, the U268 strain is an *R. redzepovicii*.

### 3.5. Comparative Genomic Analysis of Strains with Different Symbiotic Efficiency in Clover

To further analyze the genomes of the efficient *R. leguminosarum* U204 and the intermediate-efficiency strain *R. redzepovicci* U268, we performed a genomic comparison between them and the genome of the parasitic strain WSM597 obtained from GeneBank (GCF_000271785.1).

The strains WSM597 and WSM2304 (GeneBank: GCF_000021345.1) have been classified as *R. leguminosarum* bv. trifolii since their genomes were reported (Reeve et al. 2010; 2013). According to NCBI, the taxonomy checks for both strains is “inconclusive” (www.ncbi.nlm.nih.gov/assembly/GCF_000021345.1/ accessed on 15 November 2022). Our average nucleotide identity (ANI) data showed ANIb and ANIm values of 95.2% and 96.7%, respectively, only when comparing the WSM597 genome with *R. acidisoli* (Appendix A). ANI values > 96% were observed between WSM597 and WSM2304, suggesting that both strains belong to *R. acidisoli*; thus, they were identified with that species in Figure 2, Figure 3 and Figure 4. According to TYGS analysis, WSM597 and WSM2304 strains are *R. acidisoli* (Appendix A).

The three genomes of *R. leguminosarum* U204, *R. acidisoli* WSM597, and *R. redzepovicci* U268 add up to 6324 genes. From these, 4979 genes (78.7%) are common to the three strains (Figure 6). In addition, the intermediate-efficient strain U268 shares more genes with the parasitic strain WSM597 (440) than with the effective strain U204 (358; Figure 6). A limited number of genes appear to be exclusive for each genome. For instance, U204 contains 64 genes that are not present in the other two genomes, whereas the clover parasitic strain WSM597 contains 140 exclusive genes that are absent from the efficient U204 and the moderate efficient strain U268 (Figure 6). Any of these exclusive genes could be related to the symbiotic performance of the strains, such as *hrrP* and *sapA* genes.

The analysis of putative *hrrp* and *sapA* genes encoding M16 peptidases, related to the symbiotic inefficiency, showed that U204, U268, and WSM597 have four, three, and two genes encoding aminoacidic sequences for putative M16 peptidases in different contigs, respectively. Among all these putative M16 peptidases, one sequence per strain corresponded to SapA, with a chromosomal location (percentage identity > 82%, Query cover > 98% based on the chromosomal *sapA* aminoacidic sequence of *E. meliloti* 1021, GenBank ID: CAC45492.2).

The remaining aminoacidic sequences for M16 peptidases corresponded to putative HrrP peptidases. They all presented a high sequence coverage and identity percentage to the HrrP of *E. meliloti* USDA1963 (GenBank ID: AJT61688.1). Some of them were plasmidic and others chromosomic (Table 6). U268 may contain another gene encoding the M16 peptidase protein, with high identity (84%), but it is not listed in Table 6 due to a reduced coverage of 87.5% and short sequence (85 aa), which could be a pseudogene or a truncated gene.

M16 peptidases consist of two structurally related M16 domains. Whereas one is the active peptidase (M16_N; PF00675), the other is inactive (M16_C; PF05193). Using the InterPro protein families and domains database, we predicted domains in the putative HrrP of U204, U268, and WSM597 (Table 6). We found that all of them could present a domain architecture of a peptidase M16_N (PF00675) followed by two peptidases M16_C (PF05193) in a PF00675-PF05193-PF05193 architecture (Appendix A).

## 4. Discussion

### 4.1. Competitiveness of Rhizobia with Different Symbiotic Efficiencies in White Clover

Red and white clovers are used to improve pastures in Uruguay and have been inoculated with *R. leguminosarum* U204, isolated in the USA, for over the last 50 years. Although this strain is efficient in these clovers, implantation and productivity problems have been detected one year after sowing [1,5]. These problems have been attributed to low-efficient and inefficient or parasitic rhizobia in soils, which can interfere with root nodulation and biological nitrogen fixation (BNF) [2,41]. Therefore, the clover inoculant must compete with native and naturalized rhizobia with a broad range of symbiotic efficiencies [2,3,7]. To contribute to the successful use of the current inoculant, or to the development of new inoculants, it is necessary to evaluate the nodulation competitiveness of parasitic and intermediate-efficient strains [42]. We selected strains U314, U268, and U1116, isolated from red clover nodules by Batista et al. [2], and the parasitic strain UP3 isolated from *T. polymorphum* nodules [7] to determine their efficiencies in white clover. In red clover, U268 has the same efficiency as the commercial inoculant, whereas U314 has an intermediate symbiotic efficiency, and U1116 was inefficient [2]. In contrast, we show here that the three strains U314, U268, and U1116 display intermediate symbiotic efficiencies in white clover.

Under field conditions, Irisarri et al. [3] and Batista et al. [2] showed that six months after sowing, nodules of white and red clover were occupied mainly by native or naturalized strains, despite the fact that the seeds were inoculated with the U204 strain. A similar situation occurred in New Zealand, where the diversity of rhizobia in soils can negatively affect the optimum of BNF by white clover [41]. Moreover, about 20% of native strains were more competitive and efficient than the inoculant in white clover [41]. Similarly, field experiments were carried out in other countries with different *Trifolium* spp., showing that nodule occupancies by commercial inoculants were lower than by the native strains [42,43]. However, the high competitiveness of a rhizobium is not necessarily associated with high nitrogen-fixing capacity or biomass production [44].

Even though we did not observe significant differences in plant productivity and nodule number per plant between the single inoculation of U204 and co-inoculation (test 1), the percentage of U204 nodule occupancy was lower in the presence of the U268 and U1116 strains. Interestingly, white clover biomass was not affected when the parasitic strain UP3 was co-inoculated with the U204 (Table 4). However, when we tested different mixtures of UP3 and U204, we did observe differences in biomass production and nodule numbers. In particular, we found that the parasitic effect of UP3 on the plant is only observed when UP3 is present at a much higher number than U204 (Table 5). Over time, autochthonous strains, including parasitic strains, tend to predominate above the inoculant. In this scenario, nodule occupancy by autochthonous strains can be greater than by the inoculant, affecting plant productivity. Similar results were obtained in alfalfa, whose biomass production was only affected when the parasitic:efficient strain ratio was 99:1 [45].

In controlled conditions, when seeds were co-inoculated with 99:1 of U204:UP3, neither the number of nodules nor the plant productivity declined concerning the sole inoculation of U204. Our results highlight that a persistent effort on seed inoculation is needed to improve clover crop production and mitigate the potential effect of parasitic and low-efficient strains residing in the soils. When roots already have nodules occupied by efficient rhizobia, other rhizobia strains are unlikely to form new nodules, even if the latter are at higher numbers [44].

### 4.2. Phylogenetic Relationships and Genome Comparison between Rhizobia Strains with Variable Symbiotic Efficiency

In Uruguay, the presence and diversity of clover-nodulating rhizobia with different symbiotic efficiencies have been reported [2,7]. In this study, we have investigated the phylogenetic relationship among rhizobia strains with differential symbiotic efficiencies (efficient, intermediate, and inefficient). The phylogenetic analysis of the ITS region and housekeeping genes showed that the UP3 and U1116 strains group in the same clade as WSM597 and WSM2304. Though UP3 is a strain of *R. acidisoli* [7], WSM597 and WSM2304, both parasitic of white and red clover, were isolated from Uruguay and classified as *R. leguminosarum* bv. trifolii [5,6,7]. However, according to the ITS region and housekeeping genes’ sequences analysis, we confirmed the WSM597 and WSM2304 group with *R. acidisoli* [7]. Based on genome comparison, we propose that these strains belong to the *R. acidisoli* species. Moreover, we have provided evidence suggesting that U1116, a strain previously reported with parasitic behavior in red clover [2], is an *R. acidisoli* with intermediate efficiency in white clover.

In *Bradyrhizobium* strains, phylogenetic analysis based on the HGT of symbiotic genes has contributed to understanding the adaptive nature of ineffective rhizobia strains and tracing the evolutionary origin of uncooperative rhizobia, which offer no benefit to the plant [46]. Interestingly, our phylogenetic analysis of symbiotic gene sequences suggests that U1116 acquired symbiotic genes by HGT from an efficient strain of *R. leguminosarum*, partially improving its symbiotic performance on white clover. To the best of our knowledge, this is the first report of symbiotic gene acquisition via HGT by a parasitic strain of clover, with positive consequences on symbiotic efficiency. The natural acquisition of symbiotic genes by HGT within the same species was demonstrated in different rhizobia, such as the in-situ transfer of *nodA* and *nifH* from a *Biserrula pelecinus* inoculant to other soil rhizobia [47]. Likewise, the natural acquisition of symbiotic genes by HGT was demonstrated in *Trifolium*, where native bacteria acquired symbiotic genes from efficient strains within the same species [7].

The strains U268 and U314, which displayed an intermediate symbiotic efficiency in *T. repens*, grouped together and clearly separated from the other *Rhizobium* sp. used in the phylogenetic analyses. Our results suggested that both strains could belong to a new rhizobium species. Therefore, we sequenced the genome of one of them together with the efficient strain U204, which has been used as the commercial inoculant for white and red clover in Uruguay for more than 50 years. We confirmed that U268 belongs to the new species *R. redzepovicci*, recently described by Rajnovic et al. [48]. This result is the first report of an *R. redzepovicci* strain capable of nodulating *Trifolium* sp. Regarding U204, this strain has been reported as *R. leguminosarum* bv. trifolii [2,7]. Our genome comparison confirmed that U204 is an *R. leguminosarum* strain phylogenetically related to the bv. viciae.

In summary, the genomic comparisons allowed us to determine that the efficient strain U204 belongs to *R. leguminosarum*, the parasitic strains of clover WSM2304 and WSM597 are *R. acidisoli*, and the partially efficient in white clover U268 is *R. redzepovicii*.

### 4.3. Identification of Putative hrrP and sapA Genes

Some rhizobia strains degrade NCR peptides produced by legumes, by SapA and HrrP peptidases, which play essential roles in the differentiation of rhizobia inside the nodules [10,11]. Though a low amount of NCR peptides is negative for an effective symbiosis, high doses of NCR peptides can be detrimental to rhizobia, given their toxicity [49]. Therefore, limiting NCR import, for example, through the BacA peptide importer, is critical to protecting the rhizobia against host NCR peptides [50,51]. Given the relevance of HrrP and SapA peptidases in the symbiotic establishment, and that we had rhizobia with different symbiotic efficiencies, we conducted a preliminary study on the *hrrP* and *sapA*-like genes present in the genomes of strains U204, U268, and WSM597. Even though *hrrP-*like genes have been found on accessory plasmids [10,12], we found *hrrP* genes in both chromosomic and plasmidic locations, with high query coverage and similar identity to *hrrP* from the *E. meliloti* USDA1963 strain [10]. Remarkably, we found that the genome of the inefficient strain WSM597 contains only one gene encoding a HrrP-like protein (Table 6). This gene is chromosomal and likely has orthologs in the other two genomes. In turn, the U204 and U268 strains carried two and three *hrrP*-like genes, respectively, and one per strain of these *hrrP*-like genes is plasmidic. Recent studies indicate that the effects of *hrrP* genes on rhizobia symbiotic performance are not linear but can range from strongly positive to negative, depending on the genetic background and additional factors [52]. *HrrP* genes could have a positive effect when NCR peptides reach high concentrations, and HrrP could contribute to limiting their concentration to avoid their toxic effect. In this sense, its beneficial or detrimental role could be determined by the time and quantity of its expression. For instance, *hrrp-*like genes in U268 and U204 strains could be beneficial when expressed at late stages of the symbiosis establishment, and more detrimental when expressed earlier in the interaction. Further studies are needed to determine the actual roles of these *hrrP*-like and *sapA*-like genes in the symbiotic efficiencies of the clover strains.

## Figures and Tables

**Figure 1 biology-12-00243-f001:**
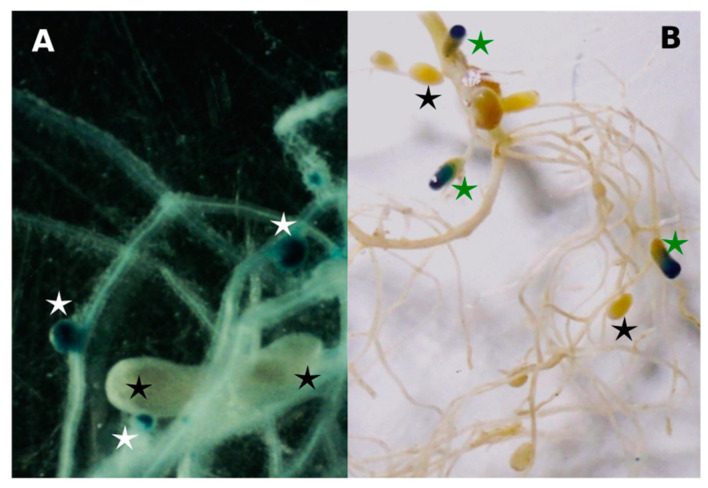
Nodule morphology. (**A**) Staining of nodulated roots of *T. repens* plants collected 21 days after seed co-inoculation. Magnification: 10X. (**B**) Staining of nodulated roots of *T. repens* plants collected at 35 dpi with U204 and UP3::*gusA*. Magnification: 3X. Nodules infected with UP3::*gusA* are indicated with white stars compared with non-stained nodules from U204 in black stars. Partially stained nodules are indicated with green stars.

**Figure 2 biology-12-00243-f002:**
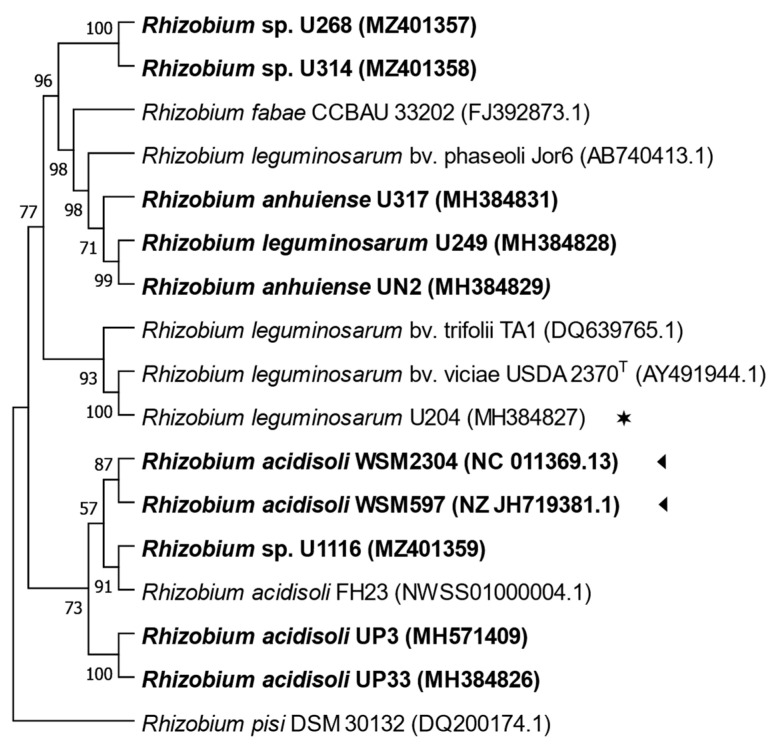
Maximum likelihood tree inferred from partial sequences alignments of ITS region (963 bp). All strains isolated from Uruguayan soils are indicated in bold letters. (◄), native strains whose genome sequences are available from Genebank; (✶), the commercial inoculant. Bootstrap values of 50 or more (based on 1000 replicates) are indicated.

**Figure 3 biology-12-00243-f003:**
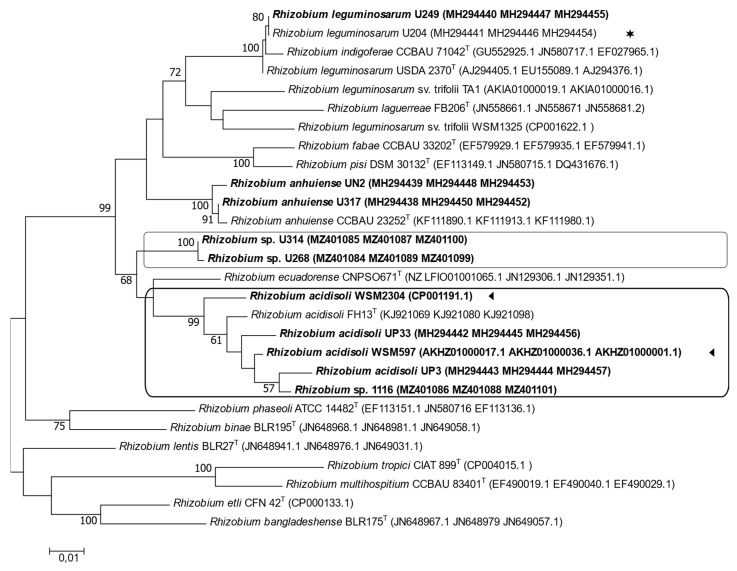
Maximum likelihood tree inferred from concatenated sequences of the housekeeping genes *atpD*, *glnII*, and *recA*. All strains isolated from Uruguayan soils are indicated in bold letters. (◄), native strains whose genome sequences are available; (✶), the commercial inoculant. Each group is identified with a box. Bootstrap values of 50 or more (based on 1000 replicates) are indicated.

**Figure 4 biology-12-00243-f004:**
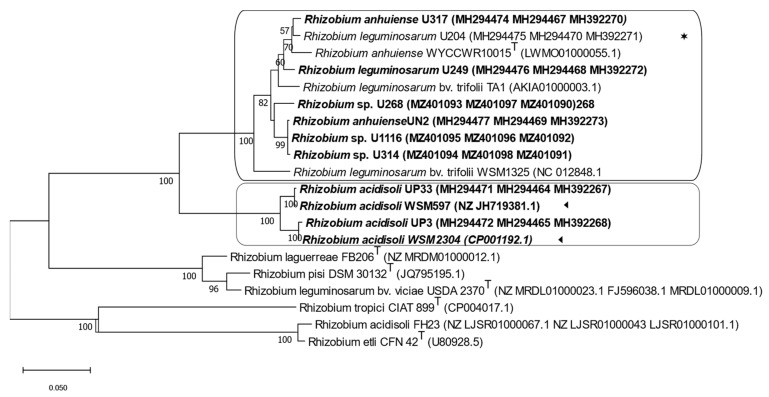
The maximum likelihood tree inferred from concatenated symbiotic genes *nodA*, *nodC*, and *nifH*. All strains isolated from Uruguayan soils are indicated in bold letters. (◄), native strains whose genome sequences are available from Genebank; (✶), the commercial inoculant. Each group is identified with a box. Bootstrap values of 50 or more (based on 1000 replicates) are indicated.

**Figure 5 biology-12-00243-f005:**
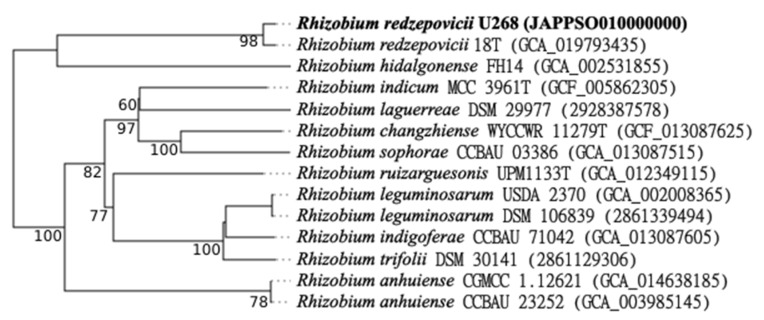
Phylogenetic analysis inferred from Genome Blast Distance Phylogeny (GBDP) distances calculated from genome sequences with TYGS (https://tygs.dsmz.de/, accessed on 7 December 2022). The branch lengths are scaled in terms of GBDP distance. The numbers above branches are GBDP pseudo-bootstrap support values > 60% from 100 replications, with an average branch support of 80.5%. The tree was rooted at the midpoint. The NCBI accession number or the IMG OID (unique numerical identifiers) are shown in brackets. Bold letters indicate the U268 strain.

**Figure 6 biology-12-00243-f006:**
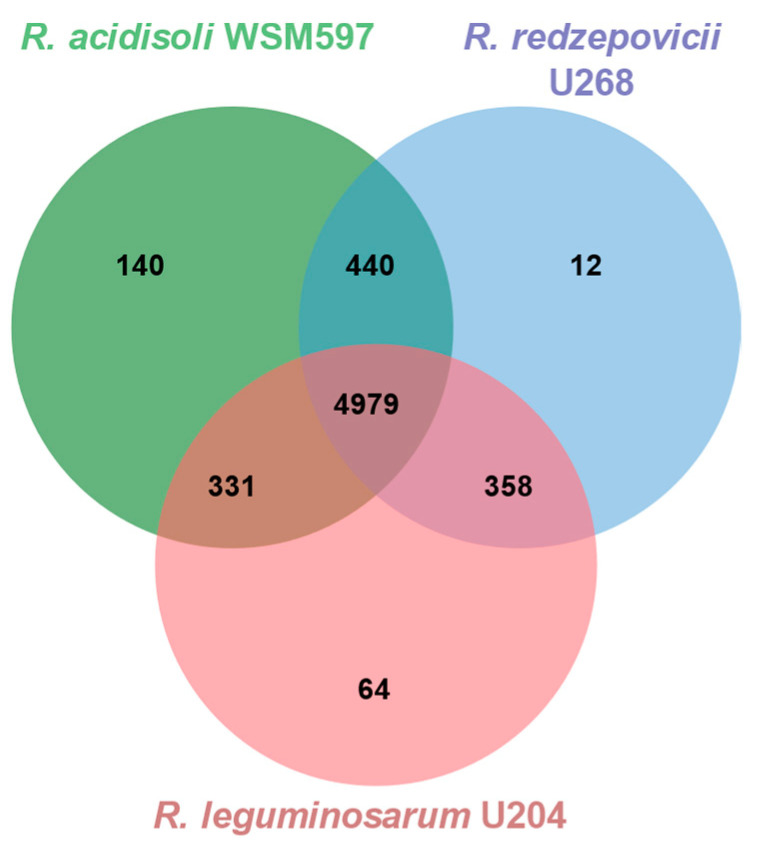
Venn diagram of homologous genes in *R. leguminosarum* U204, *R. acidisoli* WSM597, and *R*. *redzepovicii* U268. Overlapping regions represent homologous genes shared by 2 or 3 strains.

**Table 1 biology-12-00243-t001:** Rhizobia strains used in this work.

		Symbiotic Efficiency			
Strain	Original Host	Efficient	Intermediate	Inefficient	References
U314	*T. pratense*		*T. pratense*		[2]
U268	*T. pratense*		*T. pratense*		[2]
U1116	*T. pratense*			*T. pratense*	[2]
UP3	*T. polymorphum*	*T. polymorphum*		*T. repens*	[7]
U204	*Trifolium* sp.	*T. repens*, *T. pratense*		*T. polymorphum*	[1] [7]

**Table 2 biology-12-00243-t002:** Rhizobial mixtures used in nodulation competitiveness Tests 2.

Treatment	Ratio	CFU per Seed	
	UP3::*gus*A:U204	UP3::*gusA*	U204
M1	1:99	6 × 10^4^	6 × 10^6^
M2	99:1	4 × 10^6^	4 × 10^4^
M3	1:1	8 × 10^6^	8 × 10^6^
U204			8 × 10^6^
UP3::*gusA*		9 × 10^6^	

**Table 3 biology-12-00243-t003:** Symbiotic efficiency of rhizobia strains on *T. repens* in controlled conditions. Non-inoculated plants fertilized with nitrogen (+N) or non-fertilized (−N) were the controls. SDW (shoot dry weight) aerial biomass accumulated after two harvests or cuts, at 35 and 60 dpi. Means of SDW ± standard error are shown. Different letters indicate statistical differences (*p <* 0.05).

Treatments	SDW (mg/Plant)	SDW (%) Compared to +N	Symbiotic Efficiency
+N	296 ± 57 ^a^	100	
U204	268 ± 40 ^a^	91	Efficient
U268	174 ± 74 ^b^	59	Intermediate
U314	93 ± 33 ^c^	31	Intermediate
U1116	27 ± 11 ^d^	9	Intermediate
UP3	2.0 ± 1 ^e^	0.6	Parasitic
−N	2.0 ± 1 ^e^	0.6	

**Table 4 biology-12-00243-t004:** Nodulation competitiveness test 1. Nodule occupancy by strains UP3, U268, and U1116 in competition with U204::*gusA* at a ratio 1:1. Controls corresponded to seeds inoculated with each strain individually (106 CFU/seed). SDW, shoot dry weight, and the total number of nodules in each treatment at shown. Results (mean ± standard deviation) are from one of three independent experiments with a similar trend. Different letters indicate statistical differences between values within each column (*p <* 0.05).

Treatment	(%) Nodule Occupancy byU204::*gusA*	Total Nodules per Plant	SDW (mg/Plant)
U204::*gusA*/UP3	60 ^ab^	15 ± 4 ^a^	12.6 ± 3.7 ^a^
U204::*gusA*/U1116	42 ^b^	13 ± 3 ^a^	17.8 ± 8.5 ^a^
U204::*gusA*/U268	25 ^b^	11 ± 3 ^a^	17.0 ± 5.6 ^a^
U204::*gusA*	100 ^a^	12 ± 4 ^a^	18.2 ± 5.2 ^a^
UP3	0	13 ± 5 ^a^	3.5 ± 0.9 ^b^
U1116	0	10 ± 2 ^a^	10.0 ± 3.0 ^ab^
U268	0	13 ± 5 ^a^	13.0 ± 3.8 ^a^

**Table 5 biology-12-00243-t005:** **Nodulation competitiveness test 2.** Percentages of nodule occupancy and symbiotic efficiency on *T. repens* by strain UP3::*gusA*, under competitiveness experiments. Controls corresponded to seeds inoculated with UP3::*gusA* or U204 (10^6^ CFU/seed). The ratios of UP3::*gusA*:U204 in co-inoculated treatments corresponded to concentrations between 10^6^ and 10^4^ CFU/seed (Table 2). SDW, shoot dry weight. Results (mean ± standard deviation) are from one of three independent experiments with a similar trend. Different letters indicate statistical differences between values within each column (*p <* 0.05).

Treatment	UP3::*gusA*/U204 Ratios	(%) Nodule Occupancy by UP3::*gusA*	Total Nodules per Plant	SDW (mg/Plant)
M1	1:99	3 ^d^	10 ± 2 ^b^	8.3 ± 0.2 ^a^
M2	99:1	83 ^b^	14 ± 2 ^a^	3.3 ± 0.2 ^b^
M3	1:1	68 ^c^	12 ± 1 ^b^	7.3 ± 3.9 ^a^
U204	-	0 ^d^	8 ± 1 ^b^	9.7 ± 1.8 ^a^
UP3::*gusA*	-	100 ^a^	18 ± 1 ^a^	3.2± 1.0 ^b^

**Table 6 biology-12-00243-t006:** M16 peptidases identified in U204, U268, and WSM597 genomes by BlastP against HrrP aminoacidic sequence of *E. meliloti* USDA1963 (AJT61688.1) with a query coverage greater than 90%. aa: total aminoacids number.

Putative HrrP Query (aa)	Query Cover (%)	Identity (%)	Gene Location
WSM597_2 (947)	95	84.38	Chromosomic
U268_1 (858)	99	87.16	Plasmidic
U268_3 (947)	99	84.93	Chromosomic
U204_1 (911)	99	87.54	Plasmidic
U204_2 (948)	99	84.49	Chromosomic

## Data Availability

Not applicable.

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
