# Peer review of "Competitiveness and Phylogenetic Relationship of Rhizobial Strains with Different Symbiotic Efficiency in Trifolium repens: Conversion of Parasitic into Non-Parasitic Rhizobia by Natural Symbiotic Gene Transfer"

_biology, 2023, doi:10.3390/biology12020243_

Round 1
Reviewer 1 Report
It is 2.1.2. The Nodulation Competitiveness tests indicate in the text which are the M1, M2 and M3 treatments that are indicated in table 2. Because the design of the treatments is not clear at this point and although it will be understood later.
In line 146 Germany instead of Alemania
In 3.1, line 210 says shoot dry matter (SDM) but from line 211 it is indicated as SDW, possibly referring to weight, decide which notation to use.
On line 466 remove "in" which is written twice.
This work contemplates an important aspect, the analysis of the competition that may exist between the different strains of rhizobia present in soils under field conditions. I believe that his writing has the elements to be published, making a careful review of some errors found in it.
The problem mentioned in the nodulation competitiveness test could be corrected by considering the yellow and small nodules as non-functional nodules and the larger and pink ones as functional nodules. Perhaps by doing this differentiation they could obtain a better indicator of the competitiveness in each group of strains, which would even give an idea of how efficiently symbiotic nitrogen fixation could be working.
Author Response
Review 1
Comments and Suggestions for Authors
It is 2.1.2. The Nodulation Competitiveness tests indicate in the text which are the M1, M2 and M3 treatments that are indicated in table 2. Because the design of the treatments is not clear at this point and although it will be understood later.
OUR RESPONSE: We included a short clarification.
Please, see lines 140-142: Treatments M1, M2, and M3 indicated in Table 2 correspond to mixtures of UP3::gusA and U204 strains in a ratio of 99 to 1, 1 to 99, and 1 to 1 (50% of each), respectively. The final concentrations determined are shown in Table 2.
In line 146 Germany instead of Alemania
DONE. Now, line 157.
In 3.1, line 210 says shoot dry matter (SDM) but from line 211 it is indicated as SDW, possibly referring to weight, decide which notation to use.
DONE. See now line 223: shoot dry weight (SDW).
On line 466 remove "in" which is written twice.
DONE
This work contemplates an important aspect, the analysis of the competition that may exist between the different strains of rhizobia present in soils under field conditions. I believe that his writing has the elements to be published, making a careful review of some errors found in it.
The problem mentioned in the nodulation competitiveness test could be corrected by considering the yellow and small nodules as non-functional nodules and the larger and pink ones as functional nodules. Perhaps by doing this differentiation they could obtain a better indicator of the competitiveness in each group of strains, which would even give an idea of how efficiently symbiotic nitrogen fixation could be working.
OUR RESPONSE: We agree. Inefficient nodules can be differentiated from efficient ones by their coloration and shape. Still, the main problem, at least in clover, is the size of the inefficient nodules (and their color). These nodules are usually too small and white, so tiny that they cannot always be easily distinguished; they are masked by the roots, making them difficult to count. This difficulty is overcome by staining the nodules (with gusA-labeled strains). This "correction" facilitated differentiation and counting and allowed a better determination of competitiveness.

Reviewer 2 Report
The manuscript entitled “ Competitiveness and phylogenetic relationship of rhizobia strains with different symbiotic efficiency in T repens: conversion of parasitic in non-parasitic rhizobia by natural symbiotic gene transfer” deals with the comparison of different clover-specific rhizobia strains compared to a commercial strain. These results highlight that a persistent effort on seed inoculation is needed to improve clover crop production and mitigate the potential effect of parasitic and low efficient strains residing in the soils. This is an interesting topic for understanding proper productivity of white clover in Uruguay. Nevertheless, there are some comments regarding the submitted manuscript:
Minor comments
Line 2. Lack dot after T. It should be written as T. repens.
Line 21-22. “These populations include efficient rhizobia and parasitic strains that compete for nodule occu-21 pancy and prevent optimal nitrogen fixation by the grassland” This sentence is unclear. How prevent optimal nitrogen fixation by the grassland.
Line 60: If the name is used for the first time, it should be entered in its entirety. Please expand the name M. truncatula
Line 61. Please change from ‘can’ to ‘could’
Line 86: If the name is used for the first time, it should be entered in its entirety. Please expand the short name BFN
Line 90: Rhizobial strains UP3 (=P3), U1116 (=1116), U268 (=268), and U314 (=314) .. What genus do they belong to? Such information can be easily found in other publications, so it should also be provided here.
Line 368: in this place you should use only the short name BFN
Line 466. There is double in.
Author Response
Review 2
The manuscript entitled "Competitiveness and phylogenetic relationship of rhizobia strains with different symbiotic efficiency in T repens: conversion of parasitic in non-parasitic rhizobia by natural symbiotic gene transfer" deals with the comparison of different clover-specific rhizobia strains compared to a commercial strain. These results highlight that a persistent effort on seed inoculation is needed to improve clover crop production and mitigate the potential effect of parasitic and low efficient strains residing in the soils. This is an interesting topic for understanding proper productivity of white clover in Uruguay. Nevertheless, there are some comments regarding the submitted manuscript:
Minor comments
Line 2. Lack dot after T. It should be written as T. repens.
OUR RESPONSE: T. repens was changed by Trifolium repens, as requested by reviewer 3.
Line 21-22. "These populations include efficient rhizobia and parasitic strains that compete for nodule occu-21 pancy and prevent optimal nitrogen fixation by the grassland" This sentence is unclear. How prevent optimal nitrogen fixation by the grassland.
OUR RESPONSE: The sentence in lines 21-22 correspond to the Abstract in which the overall problem is presented. In the Introduction, lines 44-87 expand the topic and clarify why inefficient rhizobia stop grassland from doing nitrogen fixation. Basically, the nodule occupancy by parasitic or partially-efficient strains (like the strains cited in references 3-8) directly affects the plant host's BNF.
Line 60: If the name used for the first time, it should be entered in its entirety. Please expand the name M. truncatula
DONE
Line 61. Please change from 'can' to 'could'
DONE
Line 86: If the name is used for the first time, it should be entered in its entirety. Please expand the short name BFN
DONE. BFN was also corrected and changed by biological nitrogen fixation (BNF) in line 421.
Line 90: Rhizobial strains UP3 (=P3), U1116 (=1116), U268 (=268), and U314 (=314) .. What genus do they belong to? Such information can be easily found in other publications, so it should also be provided here.
OUR RESPONSE: Only the UP3 strain was previously identified by our group as R. acidisoli (Tartaglia et al., 2029); the other strains (U268, U1116, and U314) with intermediate efficiency were unidentified before this study.
In lines 95-96, we included the sentence: UP3 strain was previously identified in our group as R. acidisoli [8].
Line 368: in this place you should use only the short name BFN
DONE.
Line 466. There is double in.
DONE.
Review 2
The manuscript entitled "Competitiveness and phylogenetic relationship of rhizobia strains with different symbiotic efficiency in T repens: conversion of parasitic in non-parasitic rhizobia by natural symbiotic gene transfer" deals with the comparison of different clover-specific rhizobia strains compared to a commercial strain. These results highlight that a persistent effort on seed inoculation is needed to improve clover crop production and mitigate the potential effect of parasitic and low efficient strains residing in the soils. This is an interesting topic for understanding proper productivity of white clover in Uruguay. Nevertheless, there are some comments regarding the submitted manuscript:
Minor comments
Line 2. Lack dot after T. It should be written as T. repens.
OUR RESPONSE: T. repens was changed by Trifolium repens, as requested by reviewer 3.
Line 21-22. "These populations include efficient rhizobia and parasitic strains that compete for nodule occu-21 pancy and prevent optimal nitrogen fixation by the grassland" This sentence is unclear. How prevent optimal nitrogen fixation by the grassland.
OUR RESPONSE: The sentence in lines 21-22 correspond to the Abstract in which the overall problem is presented. In the Introduction, lines 44-87 expand the topic and clarify why inefficient rhizobia stop grassland from doing nitrogen fixation. Basically, the nodule occupancy by parasitic or partially-efficient strains (like the strains cited in references 3-8) directly affects the plant host's BNF.
Line 60: If the name used for the first time, it should be entered in its entirety. Please expand the name M. truncatula
DONE
Line 61. Please change from 'can' to 'could'
DONE
Line 86: If the name is used for the first time, it should be entered in its entirety. Please expand the short name BFN
DONE. BFN was also corrected and changed by biological nitrogen fixation (BNF) in line 421.
Line 90: Rhizobial strains UP3 (=P3), U1116 (=1116), U268 (=268), and U314 (=314) .. What genus do they belong to? Such information can be easily found in other publications, so it should also be provided here.
OUR RESPONSE: Only the UP3 strain was previously identified by our group as R. acidisoli (Tartaglia et al., 2029); the other strains (U268, U1116, and U314) with intermediate efficiency were unidentified before this study.
In lines 95-96, we included the sentence: UP3 strain was previously identified in our group as R. acidisoli [8].
Line 368: in this place you should use only the short name BFN
DONE.
Line 466. There is double in.
DONE.

Reviewer 3 Report
In this paper, the authors report on the nodulation competitiveness and symbiotic nitrogen fixation ability of 4 rhizobial strains interacting with Trifolium repens. They further sequence 2 complete genomes, including that of the commercial inoculant U204. They describe several interesting results:
- The high competitiveness for nodulation of 2 strains with intermediate nitrogen fixation abilities
- The first identification of Rhizobium redzepovicii as a symbiont of T. repens
- The presence of chromosomally-encoded hrrP genes
- Some evidence supporting the horizontal-transfer of symbiotic genes into a so-called ‘parasitic’ strain that may allow intermediate fixation levels
Overall, it is an interesting manuscript that makes a valuable contribution to the field.
Major comments:
- Young et al. Genes 12:111 (2021) (https://doi.org/10.3390/genes12010111) recently published an update on the phylogeny of R. leguminosarum species complex. In particular, they identify 18 genospecies within this species complex. The authors should make use of this resource to identify the genospecies to which strain U204 belongs.
- In tests 1 and 2, the presence of the gusA gene may alter nodulation competitiveness of the tagged strains. Indeed, in reference 4 (Irisarri et al. 2019), U204::gusA seems to be much less competitive than U204. It would be useful to have this additional control (e.g. U204::gusA vs. U204) in the experiments reported in this manuscript, or at least discuss the results in the light of the
Minor comments:
- L. 2: Write “Trifolium repens” instead of “T repens”?
- L. 146: Replace “Alemania” by “Germany”
- L. 203-204: The LSD post-hoc test was used, but the author should precise which test was used to determine if there was at least one significant difference in the tested values (probably an ANOVA?)
- L. 210: “Shoot dry matter” is abbreviated as “SDM”, but SDW is used in the rest of manuscript
- L. 238: “Table 5”: do the authors mean “Table 4”?
- L. 252-258 (Table 5): in test 2, the authors report a SDW of 9.7 mg/plant with U204, while they observed a SDW of 18.2 mg/plant with U204::gusA in test 1. Could the authors comment on this big difference? Is it representative of the biological variation observed between experiments? Or do they suspect an effect of the gusA gene?
- L. 267: “and mostly localized on the mail root” should be removed.
- L. 270 (Figure 1): the picture in Figure 1B is not very clear, it is hard to visualize individual nodules as they overlap with numerous root sections. Could the authors show a better picture, with less roots and more discernable nodules?
- L. 382-384 (Table 6): Are there some additional copies (located on plasmids or chromosomes) of hrrP in E. meliloti USDA1963? Did the authors did a blast of this sequence in the whole genome of strain USDA1963?
- L. 410-412: “A similar situation occurred in New Zealand, where the diversity of rhizobia in soils can negatively affect the optimum of BNF in white clover.” Cite a reference supporting this sentence.
- L. 626-628 and L. 631-632: References 40 and 42 seem inappropriate (maybe an issue with first author’s names being the same as in the intended reference?).
- The names of strains should be consistent throughout the manuscript: check for example “P3::gusA” (L. 134), “U204 and 268” (L. 182), “Rhizobium sp. 1116” (Figure 3).
Author Response
Review 3
Comments and Suggestions for Authors
In this paper, the authors report on the nodulation competitiveness and symbiotic nitrogen fixation ability of 4 rhizobial strains interacting with Trifolium repens. They further sequence 2 complete genomes, including that of the commercial inoculant U204. They describe several interesting results:
- The high competitiveness for nodulation of 2 strains with intermediate nitrogen fixation abilities.
- The first identification of Rhizobium redzepovicii as a symbiont of T. repens
- The presence of chromosomally-encoded hrrP genes
- Some evidence supporting the horizontal-transfer of symbiotic genes into a so-called 'parasitic' strain that may allow intermediate fixation levels.
Overall, it is an interesting manuscript that makes a valuable contribution to the field.
Major comments:
- Young et al. Genes 12:111 (2021) (https://doi.org/10.3390/genes12010111) recently published an update on the phylogeny of R. leguminosarum species complex. In particular, they identify 18 genospecies within this species complex. The authors should make use of this resource to identify the genospecies to which strain U204 belongs.
OUR RESPONSE: We used the Type Genome Server (TYGS) for a whole genome-based taxonomic analysis using a representative genome for each genospecies group reported by Young et al. 2021. The phylogenetic analysis was included in the Supplementary material, Figure S6.
Lines 341-343, we included the following: Among the 18 distinct genospecies groups by Young et al. [53]) for the Rhizobium leguminosarum species complex, the strain U204 belongs to the genospecies group E: R. leguminosarum (Fig. S6).
The reference for Young et al. 2021 was also included; see [53].
- In tests 1 and 2, the presence of the gusA gene may alter nodulation competitiveness of the tagged strains. Indeed, in reference 4 (Irisarri et al. 2019), U204::gusA seems to be much less competitive than U204. It would be useful to have this additional control (e.g. U204::gusA vs. U204) in the experiments reported in this manuscript, or at least discuss the results in the light of the
OUR RESPONSE: in Irisarri et al. (2019), our group reported the nodulation kinetics in white clover, and there was no difference in the time to first nodule appearance or in nodulation rate between U204 and U204::gusA.
The nodulation kinetic of UP3 and UP3::gusA (parental and tagged strains, respectively) was done in this study, and it was already mentioned in Lines 129-133.
Ines 129-133: The nodulation kinetics and symbiotic efficiency of a few transconjugants were evaluated in T. repens, as described previously [4]. UP3::gusA and U204::gusA, showing a symbiotic behavior similar to the untagged, wild types UP3 and U204, were selected for the nodulation competitiveness assays under controlled conditions.
Minor comments:
- L. 2: Write "Trifolium repens" instead of "T repens"?
DONE
- L. 146: Replace "Alemania" by "Germany"
DONE
- L. 203-204: The LSD post-hoc test was used, but the author should precise which test was used to determine if there was at least one significant difference in the tested values (probably an ANOVA?)
DONE. See now lines 215- Posthoc pairwise comparison was analyzed from plants assays based on LSD (least significant difference) with ANOVA to explore differences between means.
- L. 210: "Shoot dry matter" is abbreviated as "SDM", but SDW is used in the rest of manuscript
DONE. See now line 223: shoot dry weight (SDW).
- L. 238: "Table 5": do the authors mean "Table 4"?
DONE. Indeed, now line 253- now: Table 4.
- L. 252-258 (Table 5): in test 2, the authors report a SDW of 9.7 mg/plant with U204, while they observed a SDW of 18.2 mg/plant with U204::gusA in test 1. Could the authors comment on this big difference? Is it representative of the biological variation observed between experiments? Or do they suspect an effect of the gusA gene?
OUR RESPONSE: U204::gusA clone was compared with its parental strain concerning the nodule number and nodulation rate in white clover without differences. As the reviewer mentioned, the SDW varies typically between different plant assays. Herein, we present the results of one of three independent plant experiments; the three showed similar results with a variation in SDW between 7.34 and 18.2 mg per plant for U204/U204::gusA treatment 21 days after inoculation.
- L. 271: "and mostly localized on the mail root" should be removed.
DONE
- L. 270 (Figure 1): the picture in Figure 1B is not very clear, it is hard to visualize individual nodules as they overlap with numerous root sections. Could the authors show a better picture, with less roots and more discernable nodules?
OUR RESPONSE: Figure 1B was changed by a better one in which several partially stained nodules are clearly visible.
We also included the following (lines 286-288): Some mixed nodules (partially stained by strain UP3::gusA) are observed on roots (Fig. 1B). In the field, the presence of mixed nodules is relatively abundant (data not shown).
- L. 382-384 (Table 6): Are there some additional copies (located on plasmids or chromosomes) of hrrP in E. meliloti USDA1963? Did the authors did a blast of this sequence in the whole genome of strain USDA1963?
OUR RESPONSE: To the best of our knowledge E. meliloti USDA1963 has only one copy of hrrP gene with a plasmidic location. We did not find a genome sequence available for this strain in NCBI databases.
- 410-412: "A similar situation occurred in New Zealand, where the diversity of rhizobia in soils can negatively affect the optimum of BNF in white clover." Cite a reference supporting this sentence.
DONE. The reference [42] was added.
- L. 626-628 and L. 631-632: References 40 and 42 seem inappropriate (maybe an issue with first author's names being the same as in the intended reference?).
OUR RESPONSE: We thank the reviewer for finding these errors. Please see the correct references in Lines 659 and 668.
- The names of strains should be consistent throughout the manuscript: check for example "P3::gusA" (L. 134), "U204 and 268" (L. 182), "Rhizobium sp. 1116" (Figure 3).
OUR RESPONSE: The whole text was checked, and the strain's names were standardized.
